# Motivational Interviewing and Childhood Caries: A Randomised Controlled Trial

**DOI:** 10.3390/ijerph20054239

**Published:** 2023-02-27

**Authors:** Peter Arrow, Joseph Raheb, Rowena McInnes

**Affiliations:** 1Dental Health Services, Department of Health, Perth, WA 6152, Australia; 2Dental School, University of Western Australia, Crawley, WA 6009, Australia; 3Australian Research Centre for Population Oral Health, University of Adelaide, Adelaide, SA 5005, Australia; 4Faculty of Health Sciences, Curtin University, Bentley, WA 6102, Australia

**Keywords:** anticipatory guidance, caries prevention, early childhood caries, motivational interviewing, self-efficacy

## Abstract

Background: This study tested the occurrence of early childhood caries (ECC) and changes in potential mediators of ECC after an early childhood oral health promotion intervention. Methods: Consenting parent/child dyads in Western Australia were randomised into test [motivational interviewing (MI) + anticipatory guidance (AG)] or control (lift the lip assessments by child health nurses). A questionnaire at baseline and follow-ups (at 18, 36 and 60 months) evaluated the parental factors and the children clinically examined. Data were analysed using parametric and non-parametric tests for two groups and paired comparisons. Multivariable analysis used negative binomial regression with robust standard errors for over-dispersed count data and effect estimates presented as incidence rate ratios. Results: Nine hundred and seventeen parent/child dyads were randomised (test *n* = 456; control *n* = 461). The parental attitude toward a child’s oral hygiene needs improved among the test group at the first follow-up (*n* = 377; baseline 1.8, SD 2.2, follow-up 1.5, SD 1.9, *p* = 0.005). Living in a non-fluoridated area and parents holding a fatalistic belief increased the risk of caries (IRR = 4.2, 95% CI 1.8–10.2 and IRR = 3.5, 95% CI 1.7–7.3), respectively, but MI/AG did not reduce the incidence of dental caries. Conclusion: The brief MI/AG oral health promotion intervention improved parental attitude but did not reduce ECC.

## 1. Introduction

In Western Australia (WA), and elsewhere, early childhood caries (ECC) is a significant public health concern. Among 5-year-old West Australian children seen within the WA School Dental Service (SDS), one in three has already experienced dental decay of the primary teeth, the majority of which is largely untreated [1]. While in Australia, 26% of 5–6-year-old children have untreated dental decay in their primary teeth [2]. The management of dental decay in this young age group is demanding and often necessitates care under general anaesthesia, which is not without risk, and is expensive [3,4]. ECC also has significant negative impacts on the family and the quality of life of the child [5,6,7].

The conceptual model as outlined by Fisher-Owens, which posits the influences at the child-, family-, and community-level, points to opportunities for interventions to affect ECC [8]. Various educationally-based approaches targeted at the family-level influences to reduce the occurrence of ECC have been tested, and systematic reviews have indicated the low level of evidence available for the effectiveness of dental health education programs in reducing ECC [9,10]. Interventions using the motivational interviewing (MI) approach aimed at motivating parents to change their behaviour to improve childhood oral health appears to show promise [11]. MI is an empathic counselling approach which empowers the participant through exploring and resolving ambivalence and supporting self-efficacy for behaviour change [12].

Variants of the MI counselling approach have been tested with mixed results; some studies found positive impacts on parental attitude and beliefs on ECC [13], and reduction in ECC [14,15,16] while others found limited or no impact on ECC [17,18]. Recent systematic reviews have suggested that the MI approach, whilst showing promise, requires further investigation [19,20,21,22].

Anticipatory guidance (AG) which seeks to provide practical information appropriate for the developmental stage of the child to parents in anticipation of significant milestones for the child has shown positive results in the prevention of early childhood decay in an Australian setting [23]. The effectiveness of MI might be enhanced when it’s combined with other interventions and a combined AG and MI approach may have additive effects [24].

Psychosocial factors including parental self-efficacy, fatalistic attitudes, and stress are potential mediators that were expected to impact ECC [25,26]. It has been suggested that higher levels of parental self-efficacy might improve oral health by affecting oral health-promoting behaviours among children [27]. In a cross-sectional study, parental self-efficacy has been reported to be associated with a child’s toothbrushing behaviour with a higher frequency of brushing, while parental stress levels had no significant effect on brushing behaviour [28].

The Fisher-Owens model suggests that changes in the child’s oral health by interventions targeted at the parent are likely to be mediated through changes in parent factors. The aim of this study was to further evaluate the effectiveness of the MI/AG approach in reducing the occurrence of ECC using a randomised controlled study design. The study tested whether the MI/AG approach improved parental attitudes and beliefs about ECC and reduced the occurrence of ECC compared with the standard approach being delivered in WA, whilst also testing the potential mediated effects of parental psycho-social factors on ECC.

## 2. Materials and Methods

The details of the study design, methods and sample size estimates have been reported [1]. Briefly, the study was a two-arm, parallel-group, randomised controlled trial set within metropolitan Perth, the capital city of WA (fluoridated at 0.8 mg/L fluoride) and the regional cities of Bunbury/Busselton (non-fluoridated, 0.2 mg/L fluoride). Approximately 514 children in each arm of the trial were planned to be recruited. Allowing for loss to follow-up over the follow-up period, an initial sample of approximately 750 in each arm of the trial was required.

*Study sample*—In WA, an early childhood health contact system exists where parents of newborn children are contacted by the local child health nurse within the first few weeks after a child’s birth to help support the mother and undertake health checks. Parent/child dyads (child year of birth 2011–2013) who attended the early childhood health checks at the local child health centres when the child was 6–12 weeks old were invited to participate. Those consenting were stratified into Perth metropolitan area (fluoridated) and regional area (non-fluoridated), and randomly allocated using computer-generated randomisation blocks. Study participant pathway through the study is shown in Figure 1.

*Study variables*—The primary outcome of interest was the incidence of dental caries. The follow-ups were scheduled for evaluation when the children were expected to be about 18 months of age, 3 years of age and 5 years of age using the ICDAS criteria [29]. The examinations were undertaken by two calibrated examiners, blind to group allocation status (but aware of exposure to fluoridated water), in a knee-to-knee position under standard dental lighting in a dental clinic. Modified ICDAS criteria were used due to the age of the child; teeth were dried with cotton wool/gauze instead of air-drying and ICDAS code 1 was not scored. Caries experience was expressed as a count of tooth surfaces that were scored as decayed (ICDAS code 3–6), missing, filled surfaces (dmfs). Surfaces with a white spot lesion (ICDAS code 2) were not counted. Caries prevalence was expressed as the proportion of children with dmfs > 0. Oral hygiene was determined using the Silness-Löe index through visual assessment of the presence of plaque on index teeth (55, 65, 75, 85) at the second and third follow-ups. Plaque extent was coded at four levels; 0 = no plaque, 1 = thin film of plaque which can be seen by scraping with a periodontal probe, 2 = moderate amount of plaque visible with the naked eye, 3 = abundance of plaque, and an overall score obtained by summing the scores of the four teeth [30]. The examiners were calibrated prior to the fieldwork (inter-examiner kappa = 0.89; intra-examiner kappa = 0.97 and 0.89). Time constraints during fieldwork, which included anthropometric measures taken on the child and the parent, prevented a re-examination of the study participants for inter- and intra-examiner reliability evaluation during the fieldwork phase.

The secondary outcome was the impact of the intervention on parental factors. Information on parental factors (beliefs, attitudes, stress, self-efficacy, social support, Appendix A) expected to mediate the incidence of ECC and socio-demographic factors (mother’s education and family income) were collected via a self-completed questionnaire at baseline and at follow-ups [26]. The parent’s education level was categorised into high school or less (low), technical college/trade (medium) and university (high). Family income was categorised into low < $80,000 per annum and high ≥ $80,000 per annum. The questionnaire used validated measures on parental beliefs and attitudes towards childhood oral health, the extent of oral health fatalism, self-efficacy, stress levels and availability of instrumental support, Appendix A [25,31].

The scales were scored by summing the values of each question in the scale. Higher scores indicate poorer knowledge and more negative attitudes towards early childhood oral health, greater self-efficacy, and lower parental stress. Change scores in parental attitude from baseline to follow-up were obtained by subtracting follow-up scores from baseline scores. Positive results mean an improvement in parent attitude toward “baby bottle use” and “child oral hygiene needs”. Changes in parental self-efficacy and stress levels were obtained by subtracting baseline scores from follow-up scores; positive results mean improved self-efficacy and lower stress levels.

*Counsellor training*—Six oral health counsellors were recruited and trained over a two-day workshop comprising didactic presentations, role-playing exercises and practical sessions in MI and AG. Evaluation of the training showed an increase in the knowledge of MI principles and practices among the trainees [32]. Counsellors were supported throughout the intervention period by the researchers providing feedback at three one-day follow-up meetings. Four counsellors participated in the counselling sessions at commencement, but one withdrew and was replaced with another trained counsellor.

*The intervention*—Trained counsellors contacted the study participants within 4 weeks of consent for those allocated to the intervention arm of the study for counselling sessions. The intervention comprised three counselling sessions within a 6-month period; the first session was a face-to-face meeting at the participant’s home and the subsequent two sessions were negotiated between the counsellor and the participant, and conducted either face-to-face or via a telephone. The sessions followed a structured approach, modelled on that developed by Weinstein [11]:rapport establishment and identification of oral health and nutritional needs using empathic reflective listening;presentation of a menu of options and information with permission;discussion of options and elicitation of “change talk”;elicitation of parental importance and confidence in behavioural change;the development of a behaviour change plan; anda schedule of follow-up.

All discussions were conducted in a collaborative, person-centred approach of the MI. The menu items presented a range of topics which had the potential to affect early childhood dental decay. The discussion was limited to one or two topics of primary concern to the study participant. Anticipatory guidance was also provided and followed the anticipated milestones of teething and transition from wholly breast- or formula-fed to solid foods [33]. Children in the intervention group received counselling sessions in addition to the standard care available to the control children.

*Control*—The control group was provided with the standard care delivered through the universal “Lift the lip” program (available throughout WA since 2011). Children were screened, usually by lifting the upper lip and inspecting the upper anterior teeth for signs of ECC, by a child/community health nurse when the child presents for a health review at 8, 18 and 36 months of age. If signs of decay are detected, the parent is offered a referral to a dental practitioner for care. General anticipatory guidance would also be provided for age-specific milestones in growth and development.

*Statistical analysis*—Intention to treat analysis was undertaken, however, a few participants were excluded from analysis after randomisation due to the unavailability of contact details to enable the delivery of the intervention. Changes from baseline to follow-up were tested using parametric and non-parametric paired analyses (paired *t*-test and Wilcoxon matched-pairs signed-rank test), and differences between the test and control groups were tested using unpaired parametric and non-parametric analyses (*t*-test and Wilcoxon rank-sum test). Parental support was compared between groups using the Chi-squared statistic and changes from baseline to follow-up were tested using the matched pairs chi-square (McNemar’s Chi-square). Caries outcomes, expressed as caries experience (dmfs) between the groups were tested using non-parametric analyses due to the highly skewed non-normal data (Wilcoxon rank-sum test). Univariable negative binomial regressions were undertaken of the caries experience expressed as decayed, missing and filled surfaces against the socio-demographic and psychosocial factors and the MI intervention. Variables with *p* values ≤ 0.35 were included in the multivariable negative binomial regression model with robust standard errors [34]. The final model with the group allocation with selected variables from the univariable analysis is presented. The coefficients of the negative binomial regression are presented as rate ratios. All analyses were undertaken on a personal computer using the statistical package STATA 17 [35].

## 3. Results

The flow of study participants is shown in Figure 1. The number of completed questionnaires and the number of children examined differed because the questionnaires were mailed to the participants for completion before attending the clinical examination appointment and some parents failed to attend the appointment. Also, some of whom attended had not completed the questionnaire and requested to complete the questionnaire at home rather than at the examination, but failed to submit the questionnaire.

Participant recruitment commenced in July 2011 and was completed in December 2013. The first follow-up was from November 2012–August 2015; the second follow-up January 2014–July 2017; the third follow-up July 2016–November 2018. Of the 917 parents who consented to participate and were randomised, 54 did not provide contact details and were unable to be contacted for further follow-ups and were excluded from follow-up analyses. At the final follow-up, 69% of those successfully allocated completed the questionnaire and 66% were examined. Five hundred and fifty-six children were examined at all follow-ups. The majority of the questionnaires were completed by the mother of the child (98%). The mean intervals from baseline to the first follow-up, the second follow-up and third follow-up examinations were 1.7 years (SD 0.7), 3.5 years (SD 0.8), and 5.0 years (SD 0.6), and the mean age at follow-up was 2.0 years (0.7), 3.7 years (0.8). and 5.3 years (0.7), respectively. Ninety-three percent (*n* = 421) of the parents had at least one MI/AG counselling session, 87 % (*n* = 398) had at least two sessions, 64% (*n* = 289) had the three allocated sessions, and 7% (*n* = 34) did not receive any sessions, in spite of multiple attempts.

There were no statistically significant differences between the test and control groups in socio-demographic and psycho-social factors at baseline (Table 1), and at the follow-up (Appendix A). 

Not all items in the questionnaire were completed, hence the number of participants for each factor does not necessarily sum to the number of overall participants A higher proportion of children examined at follow-up and allocated to the control group had no social support to run errands for parents, and the difference was of marginal statistical significance ($\chi_{1}^{2}$ = 3.83, *p* = 0.05).

Characteristics of participants who were followed up and lost to follow-up are shown in Appendix A. The parents of the children examined at follow-up had a higher educational level, a better attitude toward the use of a baby bottle, higher self-efficacy, and more parental support for childminding, loan of money and transport assistance than parents of children who were lost to follow-up.

Analysis of changes in potential psycho-social mediators of ECC from baseline to the first follow-up is shown in Table 2. Test group parents’ attitude toward “baby oral hygiene needs” improved at follow-up, while parents in both groups experienced a reduction in self-efficacy and an increase in stress without any change in attitude towards oral health fatalism.

Caries prevalence and experience of participants and plaque levels at each follow-up are shown in Table 3. The overall caries experience of the study sample was 0.23 dmfs and 0.12 dmft, at the second follow-up and 0.88 dmfs and 0.45 dmft at the final follow-up, and the proportion of children with dmfs > 0 was 14.8% at the final follow-up. The differences in caries prevalence and experience between the two groups at the follow-ups were not statistically significant. Many children had unerupted index teeth for plaque scoring at the first follow-up and the plaque score is not presented. Both groups increased in the extent of plaque accumulation at the 2nd and 3rd follow-ups, Wilcoxon matched-pairs signed-rank test, *p* < 0.001. The test group had slightly lower plaque scores at each follow-up, but the difference between the groups was not statistically significant.

Univariable negative binomial regression found a 50% increased risk of caries experience among control children, although it was not statistically significant. There was also a near 3-fold, statistically significant increased risk of caries among children whose parents, at baseline, held a fatalistic attitude about a child experiencing dental caries, IRR = 2.9 (95% CI 1.26–6.68) Table 4. Children in high-income households and whose parents had more than year 12 level education were relatively protected from dental caries, IRR = 0.53 (95% CI 0.26–1.07) and IRR = 0.64 (95% CI 0.22–1.90) IRR = 0.50 (95% CI 0.18–1.36), respectively. 

Multivariable negative binomial regression modelling found a similar level of non-statistically significant increased risk of caries among the control group. Also, a child living in a non-fluoridated, regional community had a 4-fold increased risk of dental caries compared with children living in fluoridated, metropolitan community, Table 5. Children whose parents were fatalistic about their child experiencing dental caries were at a 3.5-times higher risk of caries than children whose parents were non-fatalistic. Older children were at an increased risk of dental caries experience while children whose parents were tertiary educated were relatively protected from dental caries compared with parents who had technical/trade qualifications or year 12 or less high-school education.

## 4. Discussion

This study tested an oral health promotion intervention comprising motivational interviewing and anticipatory guidance (MI/AG) among parents of newborn infants and found little impact on caries incidence and caries experience. The combined approach reflects the ‘real world’ application of oral health promotion to parents/carers of newborn infants where anticipatory guidance would be routinely used to indicate age-specific milestones in growth and development. However, a combined intervention means that the impact of the individual components could not be isolated.

The study participants were not a random sample of the West Australian population, even though about 90% of children aged 6–8 weeks attended the child health clinics for the universal health checks during the study period, participation in the study was voluntary and by informed consent. The participants were generally better educated and had higher family income than the general population of WA, thus, generalisation of the findings to the wider population should be undertaken with caution. The number of participants recruited for the study was less than planned, hence the power of the study might be lower than planned although univariable and multivariable analyses found statistically significant effects of possible confounding factors in the expected direction. The children in the study also had lower dental caries experience than that has been reported for the general population of five-year-old school children seen within public dental service in WA [36]. 

The study achieved a 66% retention of study participants at the final follow-up and the comparison of measured factors between the test and the control group at baseline (Table 1) and follow-up (Appendix A) suggests that randomisation was fair and no bias in the loss of participant follow-up. Higher loss of children from lower-income households; with parents who had lower education levels; and parents with a poorer attitude of “child oral hygiene needs”; and with less parental support at follow-up (Appendix A) is likely to reduce the occurrence of dental caries and reduce the generalisability of the study findings more widely, however, it was unlikely to impact on the internal validity of the study.

Parents in the MI/AG group improved their attitude toward the “oral hygiene needs” of their children to a statistically significant level from baseline, and parents in the control group had lower attitude levels at follow-up, but the difference from baseline was not statistically significant. Findings of improvements in attitude after an MI intervention have been reported [13]. However, that study was of a short 4-month follow-up and failed to consider of paired nature of before and after analysis, hence, its findings are not directly comparable to this study’s findings. Also, that study did not report on any changes in caries experience during the short follow-up period. Knowledge of oral health care practices whilst being necessary is not sufficient for caries prevention as knowledge does not necessarily translate into behaviour. In this study, although the MI/AG group improved in the knowledge of the child’s oral hygiene needs it did not translate into reduced levels of caries.

Findings of MI intervention improving oral health knowledge with little to no impact on caries experience have been reported in other studies. A 2-year follow-up cluster randomised study among pregnant women or carers of children younger than 6 years living in public housing developments in the United States reported improved knowledge over the interval among recipients of an MI intervention compared with the control group provided with fluoride varnish and health education materials, but no significant difference in caries increment nor parental self-efficacy [37]. The MI group was provided with MI intervention in addition to care provided to the control group. That study’s findings were similar to another randomised study over a 36-month observation period undertaken among American Indians in the United States, which found improved knowledge levels among those in the MI intervention group compared with the control group, who were provided with enhanced community services (oral health education brochures, toothbrush/toothpaste). The MI group received enhanced community services in addition to the MI intervention. But again, there was no statistically significant effect of the MI intervention on caries increment [38]. The authors of that study speculated that psychosocial factors (but did not report on them), such as fatalistic attitudes might influence the participants’ response to MI intervention. This study found, in multivariable analysis, that children whose parents hold a fatalistic attitude towards a child experiencing dental caries were at an increased risk of caries experience, despite improved attitude toward child oral health care measures.

The findings of this study are also similar to the findings from an intervention study among low-income families in the US which found no direct effect of the MI intervention on ECC [18]. That study did not report on the changes in baseline psychosocial factors and their association with the caries outcome. The study reported that parents with high self-efficacy undertook a greater amount of preventive healthcare behaviours, such as ensuring the child brushed their teeth twice daily and checking the child’s teeth for pre-cavitated lesions. However, the authors did not report on the potential mediating effects of self-efficacy on ECC. A subsequent causal mediation analysis of the study by Ismail et al., conducted as a methods paper on demonstrating the use of mediation analysis, did not find any mediating effects of the caregivers’ behaviour in ensuring the child brushed their teeth at bedtime on ECC. However, it was reported that the intervention had a statistically significant effect on the mediating variable [39]. In this study, there was no direct effect of the MI/AG intervention, and mediation analysis of the improvement, at follow-up, in a parental attitude of a child’s “baby bottle use” among the intervention group were not statistically significant (results not reported).

A recently reported non-randomised study undertaken in Australia in which an MI intervention was compared against a conventional caries prevention protocol, consisting of dietary analysis, oral hygiene instructions, toothbrushing techniques, appropriate fluoride toothpaste use and dietary advice, reported a superior caries preventive effect of conventional prevention compared with MI intervention [40]. The MI intervention group, although reporting higher levels of preventive behaviour (greater frequency of toothbrushing and lower frequency of consumption of sugared beverages), experienced higher caries increment over the follow-up period compared with those in the conventional prevention program. The authors suggest that because of the older age of children in their study, habits and caries risk might already be embedded, and that the reported changes in behaviours might be due to social desirability bias.

This study’s findings differ from that of another Australian study which delivered a multi-faceted oral health promotion intervention, including MI and AG alongside fluoride varnish applications among pregnant Indigenous women and their newborn infants, to prevent ECC [41]. The study found that the intervention reduced the level of dental caries experienced compared with a delayed intervention group. The authors acknowledged that the combined intervention precluded isolating the effects of the individual components. It is possible that the administration of fluoride varnish might have been a dominant preventive factor, where its use has been shown to be an effective caries preventive agent among Indigenous Australians [42].

The changes in parental self-efficacy and stress levels in the unexpected direction are not explained by the study’s findings. Anecdotal observations of the parents that attended the follow-up examination were that a considerable number had a younger infant or were expecting another child, which potentially could have led to feelings of greater stress and lowered self-efficacy. There was a positive, statistically significant correlation between changes in stress levels and changes in self-efficacy, indicating an association between higher self-efficacy levels with lower levels of stress. In this study, changes in parental self-efficacy and stress levels had little impact on carious outcomes. Parents of both groups reported lowered self-efficacy and increased stress levels from baseline at the first follow-up. However, the levels improved to baseline levels at the final follow-up. It is possible that parents were adapting and adjusting to stresses associated with parenthood at the first follow-up, whilst by the final follow-up they had adapted and self-efficacy and stress levels had returned to baseline levels. 

Multivariable analysis in this study affirms the protective benefits of exposure to community water fluoridation. Although specific measures of fluoride water exposure were not undertaken, the study participants were recruited into the study soon after birth and remained in that location throughout the study period, thus, were more likely to be exposed to the prevailing fluoride levels in the community water supply. A four-fold increased risk of caries-experience among those not exposed to community water fluoridation supports the adoption of community water fluoridation to reduce the incidence of caries.

Children of parents holding fatalistic attitudes towards the development of dental caries were at an elevated risk for caries. Similar findings have been reported in cross-sectional studies elsewhere [25,43]; children of parents with positive attitudes towards child oral health and less fatalistic attitudes had lower caries experience than children of parents with poorer attitudes towards oral health. In this study, despite the improved attitude of parents in the MI/AG group, there was little impact on the fatalistic attitude of the parents. People with fatalistic attitudes are less likely to adopt health-promoting behaviours and it is possible that the persistence of fatalistic attitudes prohibited the adoption of more preventive behaviours [44]. 

A limitation of this study is that a full evaluation of the fidelity of the MI intervention was not undertaken. An assessment of the change in knowledge and preparedness of the counsellors pre- and post-training showed good proficiency in the MI approach [32]. However, it is possible that the counsellors, who were very familiar with oral health care service delivery were not fully adherent to the MI processes during the fieldwork phase. The challenges faced by counsellors to transition to a more collaborative, guiding approach from traditional information provision and education approaches have been noted [45]. The need for high fidelity in the delivery of MI and the difficulty in achieving it has been reported and the different levels of fidelity achieved might be a reason for the divergent findings [46]. Although, studies with rigorous evaluation of MI fidelity also found little impact on caries experience and it was suggested that MI might have limited efficacy among high-risk populations [47]. However, an oral health promotion intervention which requires extensive training and monitoring to ensure fidelity of delivery might have limited applicability within large public health programs and further research is required to test its effectiveness. Also, participants in this study had higher relative incomes and were better educated than the rest of WA. Hence, their child was probably at a lower risk of dental caries and the impacts of the intervention were not as pronounced, although the MI interventions had a minimal impact when applied among some high-risk groups [18,37,40] and reducing caries incidence in others [15,41]. 

The study’s strengths are the high retention of participants over a long follow-up and the inclusion of putative psycho-social and community factors in multivariable modelling of caries outcome. The findings of strong effects of community water fluoride exposure, educational background, and fatalistic beliefs support the validity of the findings. This study’s findings of the effects of MI on ECC prevention, and the mixed findings from other studies suggest that further research is required to test the application of the MI approach in pragmatic settings for its use in caries prevention in large-scale public health settings.

## 5. Conclusions

This study found that an oral health promotion intervention using the brief MI approach and AG among parents of young children improved the parental attitude toward the oral hygiene needs of young children. However, the intervention did not reduce the incidence of dental caries. Lack of exposure to community water fluoridation and parental fatalistic attitude toward the development of dental caries increased the risk of dental caries experience while university-level parental education was protective of dental caries experience.

## Figures and Tables

**Figure 1 ijerph-20-04239-f001:**
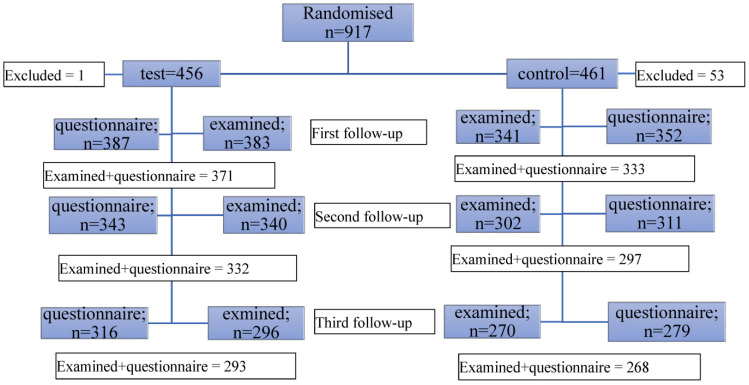
Participant Pathway.

**Table 1 ijerph-20-04239-t001:** Baseline characteristics of children in the test and control group at baseline.

Baseline Characteristic	Test	Control	*p* Value
Sex; *n* (%)			
Girls = 472 (51.4) Boys = 445 (48.6)	234 (49.6) 221 (49.7)	238 (50.4) 224 (50.3)	0.98
Location: *n* (%)			
Fluoridated = 680 (76.3) Non-fluoridated = 211 (23.7)	352 (77.4) 103 (22.6)	328 (75.2) 108 (24.8)	0.45
Education level: *n* (%)			
≤year12 = 167 (18.7) Technical/Trade = 282 (31.7) Tertiary = 442 (49.6)	78 (17.7) 135 (30.5) 229 (51.8)	89 (19.8) 147 (32.7) 213 (47.4)	0.42
Family income: *n* (%)			
<AUD$80 K = 392 (43.1) ≥AUD$80 K = 518 (56.9)	190 (48.5) 261 (50.4)	202 (51.5) 257 (49.6)	0.57
Child’s Age at baseline (months): *n* = 864 mean (SD); 3.2 (2.2)	*n* = 454 3.2 (2.1)	*n* = 410 3.4 (2.3)	0.63
Parent Baby Bottle Use Attitude: *n* = 916 mean (SD); 3.4 (3.4)	*n* = 454 3.4 (3.4)	*n* = 462 3.4 (3.3)	0.70
Parent Baby Oral Hygiene Needs Attitude: *n* = 916; mean (SD); 1.7 (2.2)	*n* = 453 1.9 (2.3)	*n* = 462 1.6 (2.1)	0.07
Fatalism:			
Fatalistic = 205 (22.5) Non-fatalistic = 708 (77.6)	104 (23.0) 348 (77.0)	101 (21.9) 360 (78.1)	0.60
Parent self-efficacy: *n* = 913 mean (SD); 18.2 (5.4)	*n* = 453 18.1 (5.4)	*n* = 460 18.3 (5.4)	0.58
Parental Stress: *n* = 916 mean (SD); 15.8 (3.8)	*n* = 454 15.9 (3.7)	*n* = 462 15.7 (3.9)	0.32
Parental Support: *n* (%): *n* = 916			
Errands			
Yes = 768 (83.8) No = 148 (16.2)	387 (85.2) 67 (14.8)	381 (82.5) 81 (17.5)	0.25
Lend money			
Yes = 792 (86.5) No = 124 (13.5)	394 (86.8) 60 (13.2)	398 (86.2) 64 (13.9)	0.78
Babysit			
Yes = 807 (88.1) No = 109 (11.9)	398 (87.7) 56 (12.3)	409 (88.5) 53 (11.5)	0.69
Lend car			
Yes = 775 (84.6) No = 141 (15.4)	387 (85.2) 67 (14.8)	388 (84.0) 74 (16.0)	0.46

**Table 2 ijerph-20-04239-t002:** Changes in psychosocial factors from baseline to first follow-up for test and control group.

Factors	Test	*p* Value	Control	*p* Value
Parent Baby Bottle Use Attitude: Baseline Followup	*n* = 381 3.2 (3.5) 3.0 (3.2)	0.20	*n* = 351 3.3 (3.4) 3.7 (3.6)	0.10
Parent Baby Oral Hygiene Needs Attitude: Baseline Followup	*n* = 377 1.8 (2.2) 1.5 (1.9)	0.005	*n* = 350 1.5 (2.0) 1.6 (2.1)	0.35
Fatalism: Non-fatalistic—Fatalistic Fatalistic—Non-fatalistic	*n* = 375 33 42 McNemar’s Chi Sq	0.30	*n* = 350 35 38 McNemar’s Chi Sq	0.73
Parent self-efficacy: Baseline Followup	*n* = 377 18.4 (5.4) 17.6 (5.5)	0.004	*n* = 346 18.5 (5.5) 17.2 (5.5)	<0.001
Parental Stress: Baseline Followup	*n* = 379 15.8 (3.8) 14.2 (3.4)	<0.001	*n* = 350 15.6 (4.0) 14.0 (3.7)	<0.001

**Table 3 ijerph-20-04239-t003:** Caries experience prevalence (*dmfs > 0) and caries experience (mean, SD *dmfs/dmft), and plaque scores at follow-ups.

	Test	Control	Statistics
First follow-up (prevalence, %, (*n*))	1.1, (4)	0.9, (3)	$\chi_{1}^{2}=0.06$ *p* = 0.81
Second follow-up (prevalence, %, (*n*))	4.7, (16)	6.3, (17)	$\chi_{1}^{2}=0.73$ *p* = 0.39
Third follow-up (prevalence, %, (*n*))	15.6, (47)	14.0, (38)	$\chi_{1}^{2}=0.28$ *p* = 0.60
First follow-up (caries experience), dmfs dmft	0.21 (2.32) 0.06 (0.66)	0.07 (1.15) 0.04 (0.57)	Mann-Whitney *p* = 0.81 *p* = 0.82
Second follow-up (caries experience), dmfs dmft	0.21 (1.86) 0.10 (0.71)	0.26 (1.64) 0.13 (0.75)	Mann-Whitney *p* = 0.36 *p* = 0.59
Third follow-up (caries experience), dmfs, dmft	0.72 (2.61) 0.42 (1.26)	1.05 (4.69) 0.48 (1.66)	Mann-Whitney *p* = 0.67 *p* = 0.55
Second follow-up (plaque score)	1.68 (2.06)	1.72 (2.16)	Mann-Whitney *p* = 0.88
Third follow-up (plaque score)	3.23 (2.23)	3.45 (2.30)	Mann-Whitney *p* = 0.29

*dmfs = d, decayed; m, missing; f, filled, surfaces; dmft = decayed, missing and filled teeth.

**Table 4 ijerph-20-04239-t004:** Univariable negative binomial regression of independent factors for the count of caries experience (dmfs, decayed, missing, filled surfaces) with robust standard errors, coefficients presented as incident rate ratios (IRR).

Independent Factor	IRR, (95% CI)	*p* Value
Group:		
Control: referent Test	1.5 (0.7–2.9)	0.27
Sex; *n* (%):		
Boys: referent Girls	0.8 (0.4–1.7)	0.62
Location:		
Fluoridated: referent Non-fluoridated	1.5 (0.8–2.9)	0.23
Education:		
≤year12: referent Technical/Trade Tertiary	0.6 (0.2–1.9) 0.5 (0.2–1.4)	0.41 0.19
Family income:		
<AUD$80 K: referent ≥AUD$80 K	0.5 (0.3–1.0)	0.07
Baseline Fatalism:		
Non-fatalistic: referent Fatalistic	2.9 (1.4–6.1)	0.01
Baseline Parental Support		
Errands:		
No: referent Yes	0.5 (0.2–1.2)	0.11
Lend money:		
No: referent Yes	0.6 (0.2–2.0)	0.38
Babysit:		
No: referent Yes	0.5 (0.1–2.2)	0.39
Lend car:		
No: referent Yes	0.6 (0.2–1.8)	0.33
Child’s Age at final follow-up	1.4 (0.8–2.3)	0.21
Change in Parent Baby Bottle Attitude	1.0 (0.9–1.0)	0.29
Change in Parent Baby Oral Hygiene Needs Attitude	1.0 (0.8–1.2)	0.87
Change in Parent self-efficacy	1.0 (0.9–1.1)	0.68
Change in Parental Stress	1.1 (1.0–1.2)	0.22

**Table 5 ijerph-20-04239-t005:** Multivariable negative binomial regression model, coefficients presented as incident rate ratios (IRR) for the count of caries experience (dmfs; decayed, missing and filled surfaces) controlling for variables with *p* < 0.35 in univariable negative binomial regression.

Variable	IRR	95% CI	*p* Value
Intervention:			
Test: referent control	1.5	0.7–3.2	0.26
Region:			
Fluoridated: referent Non-fluoridated	4.2	1.8–10.2	0.001
Income:			
<AUD$80 K: referent ≥AUD$80 K	0.8	0.4–1.7	0.64
Fatalism:			
non-fatalistic: referent fatalistic	3.5	1.7–7.3	0.001
Change in stress	1.1	1.0–1.2	0.27
Age at final follow-up	2.2	1.4–3.6	0.001
Change in baby bottle use attitude	1.0	0.9–1.1	0.53
Education:			
non-tertiary: referent tertiary	0.5	0.2–1.0	0.04
Family Support			
Do Errands:			
No: referent yes	0.5	0.1–1.7	0.27
Lend money:			
No: referent yes	1.7	0.6–5.1	0.31
Babysitting:			
No: referent yes	1.5	0.3–7.2	0.59
Transport:			
No: referent yes	0.9	0.3–3.2	0.85

## Data Availability

Data generated and/or analysed for the study are not publicly available due to privacy restrictions. Data can be made available from the corresponding author upon reasonable request.

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
