# Peer review of "Motivational Interviewing and Childhood Caries: A Randomised Controlled Trial"

_ijerph, 2023, doi:10.3390/ijerph20054239_

Round 1
Reviewer 1 Report
This manuscript tried to address one of the key dental public health problems in the Australian population.
This manuscript gives a detailed account on the conduction of RCT to achieve its desired objectives, yet occasionally it mentions extra details which are not required, similarly some of the key information is still missing (see the comments below).
Overall, this is a quality study, however following comments/suggestions may help in its improvement:
What method was used to calculate sample size?
“The study will specifically test that the MI/AG approach will improve parental knowledge and beliefs about ECC and reduce the occurrence of ECC compared with the standard approach being delivered in WA, whilst also testing the potential mediated effects of parental psycho-social factors on ECC.”
Please use past tense in the above sentences
The recruitment process took roughly 2 years (2011-13), manuscript does not reflect when exactly first follow-up started and how was data collected from the participants. Why were there different numbers of children being examined and the number of parents who filled in the questionnaire? This aspect should have been improved during the course of study (there could be less discrepancy between number of children examined and filled in questionnaire).
Manuscript should also reflect when (Month and Year) were first, second and third follow-ups completed?
“Time constraints during fieldwork, which included anthropometric measures taken on the child and the parent, prevented a re-examination of the study participants.”
Where were anthropometric measures for children and their parents used in this study? Which re-examination was not performed?
“Family income was categorised into low<$80,000 per annum and high $80,000 per annum.”
Why only low and high categories, why not atleast a medium category introduced?
During the preparation for printing of the baseline questionnaire, one item (there is not much I can do to help my child have healthy teeth) from the parental knowledge of “child oral hygiene needs” scale was inadvertently omitted. The psychometric properties of the reduced-item scale were evaluated (Cronbach’s alpha=0.80) and were considered acceptable and all subsequent questionnaires utilised the item-reduced scale to maintain consistency.
While you put so much effort in executing an RCT, this seems to be a negligible problem to be considered as a limitation and not rectified during the course of study.
How many counsellors were recruited and trained and how many of them completed the study duration. Did their loss to follow up (if there was any) had any impact on the study findings?
“Not all items in the questionnaire were completed, hence the number of participants for each factor do not necessarily sum to the number of overall participants”
What method was adopted to record parents’ responses and what were the reasons of non-completion of responses. How did it impact the findings?
Author Response
We thank the reviewer for their comments and suggestions. Please see the attached document for our response.

Reviewer 2 Report
Thank you for allowing me to review this manuscript whose purpose was to test the occurrence of early childhood caries (ECC) and changes in potential mediators of ECC after an early childhood oral health promotion intervention.
The issue the manuscript deals with is very important and intriguing. However, I have several questionable items that I ask the authors to explain and correct:
1. In my opinion, the authors did not examine knowledge (knowledge of bottle use, and knowledge of children's oral hygiene scale), but attitudes on a 5-point Likert scale. Questions with knowledge are scored as correct or incorrect, and these are opinions or attitudes. Please write the same in the title and throughout the manuscript.
2. The results tables are very confusing and difficult to follow.
In Table 3. Caries experience prevalence (*dmfs>0) and caries experience (mean, SD *dmfs/dmft), and plaque scores at follow-ups for first follow-up you have only caries experience, dmfs, where is dmft? Also where is the first follow-up for plaque for score?
In addition to the procedure, put the number of children for caries experience prevalence (*dmfs>0).
3. What does IRR mean in Table 4?
4. Are you sure that motivation interview is the right term, maybe education interview is more appropriate?
5. Please present the results in a more appropriate way, it's confusing to follow.
6. The literature is not written in accordance with the magazine's instructions.
Author Response

(The authors gave the same response as above.)

Round 2
Reviewer 1 Report
My comments have been well-addressed. I find this study of value to be included in the literature. Researchers have put much effort in executing the study as well as in the write up of this manuscript.
All the best
Reviewer 2 Report
I ask the authors to write the literature in accordance with the journal's instructions